# Biofuel from Microalgae: Sustainable Pathways

**Alvin B. Culaba [1,2], Aristotle T. Ubando [1,2,3,*], Phoebe Mae L. Ching [4,5], Wei-Hsin Chen [6,7,8] and Jo-Shu Chang [7,9,10]**

1   Mechanical Engineering Department, De La Salle University, 2401 Taft Avenue, Manila 0922, Philippines; alvin.culaba@dlsu.edu.ph
2   Center for Engineering and Sustainable Development Research, De La Salle University, 2401 Taft Avenue, Manila 0922, Philippines
3   Thermomechanical Laboratory, De La Salle University, Laguna Campus, LTI Spine Road, Laguna Blvd, Biñan, Laguna 4024, Philippines
4   Industrial Engineering Department, De La Salle University, 2401 Taft Avenue, Manila 0922, Philippines; pmlching@connect.ust.hk
5   Industrial Engineering and Decision Analytics Department, Hong Kong University of Science and Technology, Clear Water Bay, Kowloon, Hong Kong 100025, China
6   Department of Aeronautics and Astronautics, National Cheng Kung University, Tainan 701, Taiwan; chenwh@mail.ncku.edu.tw
7   Research Center for Smart Sustainable Circular Economy, Tunghai University, Taichung 407, Taiwan; changjs@mail.ncku.edu.tw
8   Department of Mechanical Engineering, National Chin-Yi University of Technology, Taichung 411, Taiwan
9   Department of Chemical and Materials Engineering, College of Engineering, Tunghai University, Taichung 407, Taiwan
10  Department of Chemical Engineering, National Cheng Kung University, Tainan 701, Taiwan
*   Correspondence: aristotle.ubando@dlsu.edu.ph

**Abstract:** As the demand for biofuels increases globally, microalgae offer a viable biomass feedstock to produce biofuel. With abundant sources of biomass in rural communities, these materials could be converted to biodiesel. Efforts are being done in order to pursue commercialization. However, its main usage is for other applications such as pharmaceutical, nutraceutical, and aquaculture, which has a high return of investment. In the last 5 decades of algal research, cultivation to genetically engineered algae have been pursued in order to push algal biofuel commercialization. This will be beneficial to society, especially if coupled with a good government policy of algal biofuels and other by-products. Algal technology is a disruptive but complementary technology that will provide sustainability with regard to the world's current issues. Commercialization of algal fuel is still a bottleneck and a challenge. Having a large production is technical feasible, but it is not economical as of now. Efforts for the cultivation and production of bio-oil are still ongoing and will continue to develop over time. The life cycle assessment methodology allows for a sustainable evaluation of the production of microalgae biomass to biodiesel.

**Keywords:** algae; biofuel production; environmental policy; life cycle assessment

## 1. Introduction

To sustain our population's growth and advancement, it is critical that renewable sources of energy be found. While it is important for energy to be produced without harmful emissions and long-term damage to the environment, it is equally important for the form of energy to be reproducible over extended periods. Biofuels are a form of renewable energy by the fact that they are generated in shorter cycles, as compared to the geological processes required to generate fossil fuels. This property

gives them two major advantages: consistency and scalability. Unlike sunlight and wind, which are subject to the variability of weather and other external forces, the volume of energy produced can be expected based on internal process parameters. The fact that these process parameters can be decided beforehand also means that the volume of production and the choice of technology are within our control.

While the concept of biofuel is promising in itself, the following questions remain: (1) what are the best sources or raw material for biofuels? (2) what are the processes involved in transforming the raw energy source into biofuel? (3) what are the optimum set-ups for implementing these processes? and finally, (4) how can biofuels replace fossil fuels? For all of the aforementioned questions, one of the key difficulties is the variety of options available. In the selection of a raw material for biofuel production, there are countless forms of biomass available. Each raw material source has its own advantages and disadvantages, and requires a different set of processes to be converted into biofuel. Furthermore, each process has its own considerations, such as the choice of equipment and operating procedures. The extensive amount of options available complicated decisions for the industry and academia alike. The research interests of both sectors require them to invest in testing and discovering effective methods of producing biofuels. The industry, in particular, faces difficulty in shifting from fossil fuels to biofuels, and potentially from inferior to superior biofuels when the availability arises.

Yet the efforts regarding biofuel have yielded some promising directions for biofuel production. In particular, microalgae have been proven to be an efficient and productive source of energy. In terms of infrastructure, they offer a higher oil yield than other biomass sources [1,2]. Table 1, containing a comparison between the productivity metrics of microalgae and those of other energy crops, shows the former's superior performance in every metric. In this sense, algae-based biofuel requires less land to produce the same amount of oil as other energy crops. This negates the need for large refineries, that would need to be situated in remote areas, resulting in high distribution costs. In terms of operational cost, algae-based biofuels can be cultivated and produced with wastewater, which was found to significantly reduce production costs when implemented [3]. These biofuels are generally easy to cultivate, not requiring any further processing aside from being kept in a growth medium [2].

**Table 1.** Comparison of biodiesel feedstock (adapted from [1]).

| Plant Source | Seed Oil Content (% Oil by Weight in Biomass) | Biodiesel Productivity (kg Biodiesel/year) | Photosynthetic Efficiency (%) | Oil yield (L/ha) | References |
|---|---|---|---|---|---|
| Corn | 44 | 152 | 0.79 | 172 | |
| Coconut | 50 | 2367 | 2.40 | 2689 | [1] |
| Sugarcane | 53 | 3696 | 2.24–2.59 | - | |
| Palm Oil | 36 | 4747 | 3.20 | 5950 | |
| Canola/rapeseed (*Brassica napus L.*) | 41 | 862 | - | 974 | [2] |
| Microalgae (30% oil by wt.) | 80 | 51,927 | 2.00–6.48 | 58,700 | [1] |
| Microalgae (70% oil by wt.) | 70 | 121,104 | 2.30–15.0 | 136,900 | |

While this addresses the first of the aforementioned questions (i.e., "What are the best sources or raw material for the biofuel?"), there are still the next three to consider. To address the second question ("What are the processes involved in transforming the raw energy source into biofuel?"), there are various methods of algae cultivation and oil production that may be evaluated. Various set-ups have been developed, with each seeking to introduce an improvement or a solution to a problem in algae-based biofuel production. Such problems include the trade-offs between lipid production and growth rate for algae. To address the third question ("What are the optimum set-ups for implementing these processes?"), requires the evaluation of decisions on facility planning, logistics, and co-production.

Finally, the issue of adoption (posed by the question "How can biofuels replace fossil fuels?") involves business model design, plans for expansion, and the anticipation of the consequences for society if biofuels were to replace fossil fuels to a large extent.

It is clear that, in order to produce and use algae-based biofuels in a sustainable manner, many factors would need to be considered. This problem requires a holistic perspective, promoting the complete identification of all components involved in production; able to consider all possible outcomes of a decision; and extending this evaluation to the environmental, social, and techno-economic implications of the decision. In identifying sustainable pathways for biofuel production from microalgae, this chapter will follow the approach of a Life Cycle Assessment (LCA). It is an internationally-standardized tool for evaluating the environmental performance of a single product [4]. Through its cradle-to-grave perspective, all the relevant processes are identified and accounted for. Impact assessment converts all material consumption and emission into logical implications.

This chapter is segmented into six parts. In this part, the issue of the sustainability of biofuels was decomposed into key questions, each representing a challenging aspect of biofuel production. In the second part, a collection of solutions from the academic literature, addressing each question, is presented. In the third part, the most viable solution alternatives are further evaluated from an LCA cradle-to-grave perspective, providing insights on the general direction of progress in this area. The challenges of sustaining growth and feasibility are evaluated separately from the other questions, as they follow a different timeline. The fifth part contains the practical implications of the proposed solutions, segmented by the nature of their impact as environmental, technical, economic, and social. Finally, new directions and ideas for the refinement of existing solutions are discussed in the sixth part.

## 2. Sustainable Technological Solutions

Understandably, one of the most effective ways of improving the sustainability of a product is to utilize contemporary technologies. In this part, the technologies will be divided into two major groups: those involved in algae cultivation and those involved in processing or production. The algae cultivation stage covers the growth and drying of algae. It prepares the raw materials for the state where they may be used for biofuel production. This is primarily a natural state and does not require processing, although the following sections will discuss some methods and set-ups that can influence the outcomes of cultivation. On the other hand, processing covers the phases where dried algae are converted into biofuel through chemical processes. These are primarily technology-driven processes, and the corresponding section will discuss various means of converting dried algae into energy.

### 2.1. Algae Cultivation Technology

Based on the nature of solutions, the algae cultivation stage can be further dissected into two parts: the choice of algae species to cultivate and the cultivation set-up for the algae. The former concerns the inherent properties of algae. Specific species have been identified as especially apt for lipid production. However, high lipid production is usually indicative of lower growth rates, which is a disadvantage of species with this characteristic. Table 2 shows a summary of studies on specific species of algae and the advantages that were identified with each species.

**Table 2.** Summary of algae species characteristics.

| Algae Species | Advantage | Source |
|---|---|---|
| *Chlorella, Dunaliella, Chlamydomonas, Scenedesmus, Spirulina* | Appropriate for bioethanol production | [5] |
| *Scenedesmus* | Higher yield than duckweed, its feedstock has a higher heating value. | [6] |
| *Chlorella* | Yields bio-oil with low oxygen content and a comparatively high heating value when subjected to pyrolysis. | [7] |
| *Nannocholoropsis* | Yields bio-oil with lower oxygen content when subjected to pyrolism with higher amounts of the catalyst HZSM-5. | [8] |
| *Chlorococcum, Chlorella vulgaris* | Can be mass cultivated with ease using current farming technology; shows applicability in ethanol production. | [9,10] |
| *P. ellipsoidea, S. almeriensis* | High energy yield without the disadvantage of high enzyme requirements, high capital cost, and being energy intensive, as experienced with other species. | [11] |

There have also been attempts to genetically modify existing species to promote desirable qualities of algae species. In particular, there is much interest to enhance lipid production due to its association with energy yield. Microalgae are highly suitable for genetic manipulation, with its amenity for genetic transformation with foreign genes. A summary of successful genetic modifications is given in Table 3.

**Table 3.** Summary of successful genetic modifications of algae species.

| Algae Species | Modification | Impact | Source |
|---|---|---|---|
| *Madhuca indica, Balanites aegyaptiaca* | De-oiled seedcake | Fast-growing, with lesser dependence on insecticides and fertilizer. | [12] |
| *Clostridium tyrobutyricum* | Over-expressions of aldehyde/alcohol dehydrogenase | Enhanced butanol production and tolerance of butanol. | [13] |
| *Escherichia coli* | Modification of amino acid biosynthetic pathway | Higher production of 1-butanol and 1-propanol. | [14] |

Another way of improving the output from cultivation is through the very process of cultivation. At the moment, three forms of cultivation can be observed: open-culture systems (see Figure 1), closed-culture systems (see Figure 2), and hybrid two-stage cultivation. Open-culture systems are set-ups where algae are allowed to grow in open raceway ponds, subject to the conditions of the environment. This means that minimum human intervention is needed in cultivating the algae, but it also means that some yield potential is lost. This is not a problem with closed-culture systems, where the environment is controlled, yet these advantages may be offset by the higher operating and capital cost required by such facilities. Two-stage hybrid cultivation systems attempt to achieve a balance between the advantages of either system. In this case, algae are grown in enclosures, but allowed to yield in open ponds, less susceptible to yield loss from contaminants. The use of optimal cultivation techniques can simultaneously lower resource consumption (and thereby cost), while promoting a high energy yield for any species.

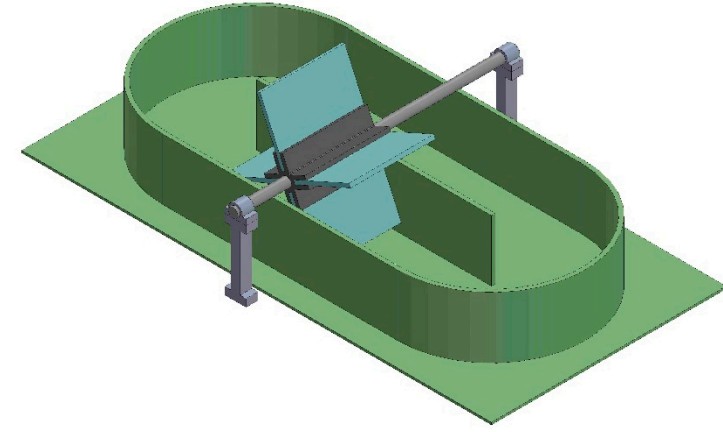

**Figure 1.** Open-culture set-up illustration.

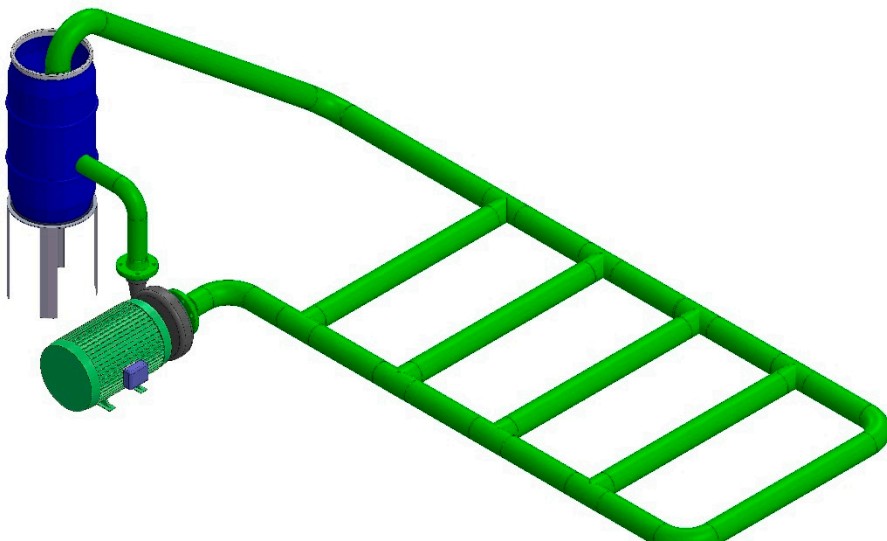

**Figure 2.** Closed-culture set-up illustration.

New approaches to the cultivation and growth of microalgae were developed and designed. Recently, an indoor hybrid helical-tubular photobioreactor was proposed by Hashemi et al. [15] for beta-carotene production Dunaliella salina cells under salt stress condition. A newly developed microgravity-capable membrane raceway photobioreactor for Chlorella vulgaris SAG 211-12 was introduced by Helisch et al. [16] for life support in space. On the other hand, Khalekuzzaman et al. [17] developed a hybrid anaerobic baffled reactor and photobioreactor for a simplified method of algal biofuel production. It is important for algal biofuel production to be simple and cost-effective to make it competitive with commercially available biofuels.

### 2.2. Biofuel Production Technology

There are also various methods that have been developed for biofuel production. Studies that develop or investigate the use of these methods are able to give insight on the optimal parameters for each process. A summary of developments in this area are given in Table 4.

**Table 4.** Summary of biofuel production processes.

| Process | Description | Recent Developments |
|---|---|---|
| **Biochemical Conversion** | Utilization of bacteria and other microorganisms in breaking down the carbohydrate content of algae into biofuels. Includes: Fermentation, anaerobic digestion | Increasing yield of feedstock, finding more efficient bacteria enzymes and more robust microorganisms. |
| **Thermochemical Conversion** | Application of high-temperature technologies to convert biomass into alcohol, hydrocarbon fuels, chemicals, and power. Includes: Gasification, pyrolysis, liquefaction | Identification of high-yielding gas mixtures for the conversion process, assessment of various algae species for aptness toward pyrolysis. |
| **Chemical Reaction (Transesterification)** | Conversion of oil into biodiesel by utilizing excess methanol. | Identification of seed oils which have a high heating value when converted into biodiesels. |
| **Direct Combustion** | Conversion of solid residue from microalgal biomass using heat and energy to produce electricity. | Studying and standardizing microalgal biomass allocation of solid residue. |

*2.3. Conversion of Microalgae to Biofuel*

There are several processes involved in converting microalgae to biofuel. These processes are the following: Cultivation, Harvesting, Drying, and Oil Extraction. Several authors have reviewed these processes over the years. [1] published an article that tackles the conversion of microalgae to biofuel. His discussion concluded that biofuel conversion will be more efficient if the cultivation process is optimized. In addition, he also made several recommendations on what type of cultivation systems to use to enhance microalgae production. Following this study, [18] reviewed the technologies for algal biofuel production, its processing, and extractions. It proposed coupling algal conversion to biofuel with carbon sequestration and wastewater treatment in order to enhance the yield and cut down its production costs. Not just in the macro side of production but also looking deeper inside the algal cell, [19] discussed the possibility of algal production via genetic alteration to produce more algal oil.

2.3.1. Cultivation and Species of Microalgae

Microalgae have various commercial applications in the areas of pharmaceuticals, cosmetics, aquaculture, nutraceuticals, and energy [20–23]. Dating back to 1939–1945, during World War II, microalgae were cultured by the military for food purposes [24,25]. In addition, as a response to environmental issues in the 1950s, algae were used to capture carbon dioxide ($CO_2$) [26]. Hence, microalgae have paved the way for addressing issues such as climate change, energy demand, and food security.

Microalgae offer higher oil yield per litter per hectare (see Table 5). Based on the table, microalgae have 100× yield of oil to convert to biodiesel. Microalgae have a lot of species discovered and undiscovered. Table 6 shows the different microalgae species and their oil content.

**Table 5.** Chemical composition of microalgae (adapted from [27]).

| Microalgae | Protein | Carbohydrates | Lipid |
|---|---|---|---|
| *Scenedesmus obliquus* | 50–56 | 10–17 | 12–14 |
| *Scenedesmus quadricauda* | 47 | - | 1.9 |
| *Scenedesmus dimorphus* | 8–18 | 21–52 | 16–40 |
| *Chlamydomonas rheinhardii* | 48 | 17 | 21 |
| *Chlorella vulgaris* | 51–58 | 12–17 | 14–22 |
| *Chlorella pyrenoidosa* | 57 | 26 | 2 |
| *Spirogyra* sp. | 6–20 | 33–64 | 11–21 |
| *Dunaliella bioculata* | 49 | 4 | 8 |
| *Dunaliella salina* | 57 | 32 | 6 |
| *Euglena gracilis* | 39–61 | 14–18 | 14–20 |
| *Prymnesium parvum* | 28–45 | 25–33 | 22–39 |
| *Tetraselmis maculata* | 52 | 15 | 3 |

**Table 6.** Oil content of microalgae (adapted from [1]).

| Microalgae | Oil Content (%Dry wt) |
|---|---|
| *Botryococcus braunili* | 25–75 |
| *Chlorella* sp. | 28–32 |
| *Crypthecodinium cohnii* | 20 |
| *Cylindrotheca* sp. | 16–37 |
| *Cylindrotheca* sp. | 23 |
| *Dunaliella primolecta* | 25–33 |
| *Isochrysis* sp. | >20 |
| *Monallanthus salina* | 20–35 |
| *Nannochloris* sp. | 31–68 |
| *Neochloris oleoabundans* | 35–54 |
| *Nitzschia* sp. | 45–47 |
| *Phaeodactylum tricornutum* | 20–30 |
| *Schizochytrium* sp. | 50–77 |
| *Tetraselmis sueica* | 15–23 |

Culturing microalgae for microalgae production was discussed in [1], which analyzed the biorefinery concept, and advances in photo-bioreactor engineering will further reduce the cost of production. This was followed by a research using an empirical and critical analysis to translate laboratory research into a full-scale commercialization application [3]. However, [28] looked at the cost, energy balance, and environmental impacts of algal biofuels and highlighted the uncertainties for microalgae production.

In culturing microalgae there are several parameters that we need to consider, such as light, $CO_2$, Temperature, pH level, and, most importantly, mixing. Microalgae are photosynthetic cells; they need light to grow. Hence, light intensity is an important aspect of growing microalgae.

There are different ways to illuminate light in the algal culture; one is with sunlight, and by using artificial ponds. These lighting systems also vary with respect to the size of the culture system that one is growing. Hence, the bigger the area that one has to cultivate, such as the open pond cultivation system, the bigger the light needed, such as sunlight. In addition, if we stick to photo-bioreactors such as flat plates, tubular, airlifts, or inclined photo-bioreactors, artificial light will suffice. There are several studies that use light as a light source (flat plate PBR and tubular PBR). Comparing both systems, flat plate PBR has higher light capture than tubular PBR, since flat plate has higher surface area than the tubular PBR [29]. There are several studies dealing with light control, since natural solar light cannot be controlled. The main focus of these studies is on PBR systems using artificial light. To give sufficient light to microalgae, the optimal range of wavelength must be between 600 and 700 nm [26].

Borowitzka [24] discussed the commercialization of the production of microalgae. He concluded that there is a need for an indoor or closed system in algal cultivation, so that there will be a controlled environment which can be more efficient in terms of yield. Hydrothermal liquefaction technology shows promising results in the production of algal biofuels, as it enables the processing of wet microalgae biomass to produce biofuels. Recent works on hydrothermal liquefaction for algal biofuels are discussed as follows. Devi and Parthiban [30] proposed the application of hydrothermal liquefaction on microalgae Nostoc ellipsosporum was cultivated in municipal wastewater to generate high bio-oil yield. Dandamudi et al. [31] applied hydrothermal liquefaction on Cyanidioschyzon merolae and Salicornia bigelovii Torr. for the production of high-valued biofuel intermediates. Arun et al. [32] used hydrothermal liquefaction on Scenedesmus obliquus to produce biohydrogen. The employment of hydrothermal liquefaction on microalgae biomass to biofuel production provides positive benefits in terms of cost-effectiveness, value-addition, and environmental impact of the algal biofuels.

### 2.3.2. Harvesting

Harvesting of algae means the detachment of algal cells to the growth medium of the cultivation system. Methods of harvesting rely on the type of algae being harvested, depending on the density, size, or even the description of the final product of the produced algae [3,33,34]. Different harvesting methods are used in algae technology. There are mechanical, chemical, biological, or even electrical means of harvesting the algae.

The harvesting process is one of the important processes in algal production in terms of cost, accounting for 30% of the overall production cost for biofuel [35]. The harvesting techniques listed are used on the application of biofuel, human and animal food, high valued products, and water quality restoration. These harvesting techniques passed all the six important criteria—biomass quantity and quality, cost, processing time, species specific, and toxicity. Coagulation/flocculation, centrifugation, and filtration are amongst the harvesting techniques that can be feasible in biofuel production [36].

### 2.3.3. Drying and Dewatering

Drying is one of the main bottlenecks in algae to biofuel production, as it accounts for around 84.9% of the total energy consumed in the whole production line [37]. There are different methods used in drying microalgae for biofuel production, such as solar dying, convective drying, rotary drying, spray drying, crossflow drying, vacuum drying, and flashing drying. Convective drying was experimented on algae with variation of temperature and wind velocity to determine the lipid yield of algae [38]. Prakash et al. [39] used a solar device to dry microalgae, specifically the spirulina and scenedesmus species. Solar drying is assumed to be an efficient drying method. However, due to weather-dependent reasons and the fact that other countries do not have enough sunlight during a 24-h cycle, the use of solar dying is a disadvantage. Moreover, when the sample ceases to dehydrate, the nutrient value in the species dissipates [40]. Hence, other forms of drying emerged, such as microwave drying of algal species [41].

### 2.3.4. Oil Extraction

Oil extraction is the process where the algal product is separated from the dried biomass. Aside from dewatering and drying, this is process also entails considerable cost and effort. Different methods, such as chemical and physical methods, are available. However, the cost of extracting the algal product is much higher than that of extracting oil from palms [1].

## 3. Applications of Sustainable Technologies

The application of the technologies discussed in the previous section are critical for meeting our Sustainable Development Goals. In particular, they are seen as a means to achieve the obsolescence of the small and portable power-generating machines being used in rural communities, which consume fossil fuels and are environmentally inefficient. As most of these rural communities have access to

biomass feedstock, which can be used in biofuel production, this could be a highly suitable alternative for their energy requirements. However, this is still dependent on the availability of the choice of technology and system design.

The number and variety of technologies available for producing algae-based biofuels call for a holistic and impartial means of comparing different methods of production. As previously mentioned, the Life Cycle Assessment (LCA) framework is one such method. A standard LCA has the following components: (1) goal and scope definition, (2) life cycle inventory, (3) impact assessment, and (4) interpretation. In goal and scope definition, a "functional unit" will be decided for the product in question. All material components and emissions generated by a product will be given in quantities of the functional unit (e.g., grams of apple fruit per box of apple juice). System boundaries will also be set, which will be the scope of the study. Although an LCA would typically entail that the scope be set from "cradle-to-grave", or raw material sourcing until the final disposal, there are some cases where it may not be practical to adopt a scope of this magnitude. In life cycle inventory, all material components and emissions are identified and quantified according to the functional unit. In this sense, the material components and emissions of a product should add up to the actual mass of the product itself. These will be converted into environmental impact in the impact assessment phase. The outcome is the impact on the quantified impact on the environment. This will be the basis for further analysis and solution development in the interpretation phase.

For biofuels, a cradle-to-biofuel scope can be adopted, starting from energy crop cultivation until biofuel production. This allows for the evaluation of the technologies discussed in the previous section. Whereas most LCA studies are typically from "cradle-to-grave", with grave indicating the disposal or any other form of expulsion from the system, this is impractical in the case of energy production, given that the energy can be used for a multitude of purposes. From an LCA, considering environmental impact alone, the most favorable alternative would be the use of natural solar power in cultivation and solar-assisted drying techniques following cultivation. Energy consumption is high under closed-culture cultivation, and the use of solar power can reduce energy consumption. However, it must be noted that the unreliability of solar power, particularly during inclement weather, may also prove to be a disadvantage.

In many cases, algal biorefineries producing biofuels alone are found to be economically infeasible, regardless of how efficient the current processing technologies are. However, the opportunity to produce co-products is a definite way of making these production systems feasible. Hydrogens, polymers, and biochemicals are just some examples of the kinds of co-products that can be produced. Some of the biofuel production processes identified in Table 4 yield co-products. Table 7 shows a summary of these co-products and their purpose in the biorefinery.

**Table 7.** Summary of co-products and purpose.

| Processes | Co-Product | Purpose |
|---|---|---|
| **Biochemical Conversion (Fermentation)** | Carbon dioxide | Can be recycled for use in cultivation. |
| **Chemical Reaction (Transesterification)** | Glycerol and methanol | Co-products can be commercialized externally. |

More comprehensive are the available biorefinery models published in literature and their corresponding co-products. These have been proven to be feasible from industrial, techno-economic, and socio-economic perspectives. A summary of the models developed is given in Table 8.

**Table 8.** Summary of biorefinery models with co-products.

| Feedstock | Main Product | Co-Product(s) | Performance Measure | Source |
|---|---|---|---|---|
| **Lignocellulosic, Macroalgae, and Microalgae** | Biodiesel and bioethanol | 1. Hydrogen 2. Polymers 3. Biochemicals | Research and development | [42] |
| **Biomass** | Biofuels | 1. Power 2. Char 3. Bio-oil 4. Gaseous fuels | Technology | [43] |
| **Biomass** | Biofuels | 1. Solvent surfactants 2. Petrol derivatives 3. Composites 4. Lubricants 5. Pastes | Industrial Metabolism | [44] |
| **Macroalgae** | Bioethanol | Energy deficit | Production | [45] |
| **Microalgae: C. reinhardtii, C. kessleri, E. gracilis, A. plantesis, S. obliquus, D. salina** | Biogas | Biohydrogen | Production | [46] |
| **Microalgae** | Electricity | 1. Biofuels 2. Omega-3 3. Algal meal 4. Biochemicals | Sustainability and energy policy | [47] |
| **Microalgae: Botryococcus braunii, Chlorella** | Petrol fraction and Biodiesel | Jet fuel | Technology evaluation | [48] |
| **Microalgae** | Biodiesel | 1. Algal meal 2. Omega-3 fatty acids 3. Glycerin | Economics and water footprint | [49] |

## 4. Case Study: A Life-Cycle-Based Production of Biodiesel from Microalgae

### 4.1. The Life Cycle Assessment (LCA) Process

LCA is a comprehensive environmental evaluation tool for holistically examining the product life cycle. It consists of four major steps: (1) the goal and scope, (2) the life cycle inventory, (3) the life cycle impact assessment, and (4) the interpretation of results. The first step is to define the goal of the LCA study through the establishment of the functional unit. After defining the goal, the next step is to decide on the scope by illustrating the system boundary of the study. An example of the system boundary is the cradle-to-grave, which indicates that the analysis includes the initial process from the raw material extraction to the preparation, the processing, the utilization, and the disposal. The system boundary also provides the degree of depth of the impacts of the indirect materials used in processing the product under study. The second step is to gather all relevant information of the input and output of the processes and the analogous environmental impact to establish the life cycle inventories. The third step is to calculate the environmental impacts and quantify the environmental burden in terms of the various environmental impact categories. The last step is the interpretation of the results, which leads to the identification of the environmental bottleneck and the recommendations to address the identified concern.

### 4.1.1. The Goal and Scope

The functional unit for the LCA case study is 1000 kg of biodiesel. This case study considers algal biomass feedstock. The processes consist of the cultivation, harvesting, drying, oil extraction, and transesterification. The generic system boundary for the microalgae-based biodiesel production is shown in Figure 3.

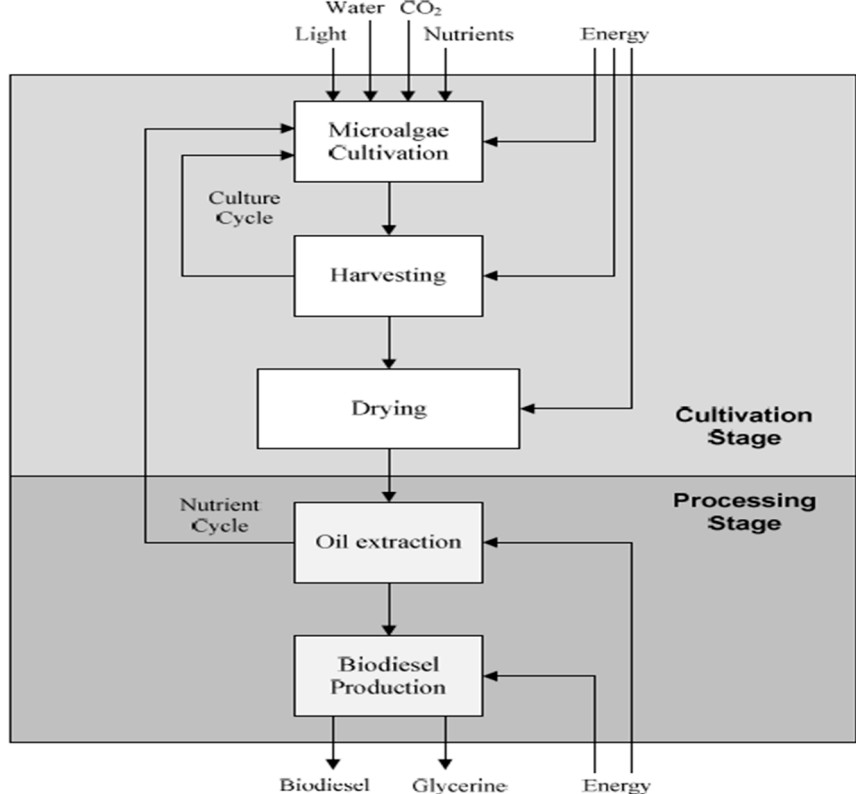

**Figure 3.** Flowchart of microalgae cultivation, harvesting, and drying to biodiesel production [3].

### 4.1.2. Life Cycle Inventory

The life cycle inventory of the microalgae biodiesel production is shown in Table 3. Table 3 presents the detailed input and output streams of the unit processes.

In Table 8, the unit processes of the microalgae biodiesel production were identified as cultivation, harvesting, drying, oil extraction, and transesterification. In order to grow the microalgae, the cultivation adopted in the study was an open pond from [50,51]. The electricity consumed for the cultivation stage is attributable to the lighting and aeration requirements for growing the biomass. After growing the microalgae, they are dewatered and harvested using a coagulation process adopted from [52]. The harvested microalgae are then dried to attain a moisture content less than 10%, using a microwave drying taken from [41]. The dried microalgae are then further processed to extract the oil, employing an oil yield of 30% based on [53]. Lastly, the extracted microalgae oil is converted to biodiesel through the conventional transesterification process adopted from [54].

In order to translate the impacts of the input and output of the unit processes of the microalgae biodiesel, the Ecoinvent LCA database from SimaPro was utilized as the source for the life cycle inventory for the study.

### 4.1.3. Life Cycle Impact Assessment

The life cycle impact assessment was performed using the Environmental Development of Industrial Products (EDIP), which employed the middle point method that accounted for the impacts as they was generated from the plant. The EDIP was utilized for the case study, as it provided a general categorization of the impact categories, enabling an easier interpretation of the results on the microalgae biodiesel production.

### 4.2. Sustainability Analysis

Microalgal biodiesel had a strong impact in 16 out of the 19 impact categories covered by EDIP. This is due to the fact that the production of biodiesel from microalgae required additional energy-intensive processes such as harvesting and drying, as shown in Figure 1. Different technologies have been explored for harvesting. A summary of these technologies is given in Table 9.

**Table 9.** Advantages and disadvantages of algae harvesting [55,56].

| Technique | Advantages | Disadvantages |
|---|---|---|
| Coagulation/flocculation | • Fast and easy technique<br>• Large scale usage<br>• Less cell damage<br>• Applied to vast range of species<br>• Less energy requirements<br>• Auto and bioflocculation may be inexpensive methods | • Chemicals may be expensive<br>• Highly pH dependent<br>• Difficult to separate the coagulant from harvested biomass<br>• Efficiency depends upon the coagulant used<br>• Culture medium recycling is limited<br>• Possibility of mineral or microbial contamination |
| Flotation | • Suitable for large scale<br>• Low cost and low space requirement<br>• Short operation time | • Needs surfactants<br>• Ozoflotation is expensive |
| Electrical-based processes | • Applicable to all microalgal species<br>• No chemical required | • Metal electrodes required<br>• High energy and equipment costs<br>• Metal contamination |
| Filtration | • High recovery efficiency<br>• Cost-effective<br>• No chemical required<br>• Low energy consumption<br>• Low shear stress<br>• Water recycles | • Slow, requires pressure or vacuum<br>• Not suitable for small algae<br>• Membrane fouling/clogging and high operational and maintenance cost<br>• High energy consumption |
| Centrifugation | • Fast and effective technique<br>• High recovery efficiency<br>• Preferred for small scale and laboratory<br>• Application to all microalgae | • Expensive technique with high energy requirement<br>• High operational and maintenance cost<br>• Appropriate for recovery of high-valued products<br>• Time consuming and too expensive for large scale<br>• Risk of cell destruction |

Since the microalgae slurry biomass is considered as a wet biomass, it requires a special method for harvesting, and in this case the coagulation method was used [52]. To successfully generate the appropriate amounts of biodiesel, the moisture content of biomass prior oil extraction should be less than, or equal to, 10% [41]. Hence, the moisture content from the harvested microalgae was still high and required further moisture removal from drying. It was found that the energy requirement per mass of dried microalgae biomass was much lower for the microwave drying compared with conventional drying methods. Due to this reason, the microwave drying was utilized as the drying method for this case study.

It was recognized that in 16 out of the 19 impact categories, the microalgae cultivation process contributed the highest impact when compared with the other processes. The cultivation stage which required lighting and aeration dominated the other processes in terms of the impact on most of the different impact categories. The environmental impact of the cultivation process can be lessened by implementing cultivation technologies which do not require lighting [57]. The drying process was observed to follow the cultivation process as the next highest process contributor to the 16 impact categories. According to [58], the impact of microalgae drying can be significantly reduced by utilizing solar-assisted and solar dryers. To further avoid the energy-intensive drying process, hydrothermal liquefaction can be employed to directly extract the lipids from microalgae [59]. Recently, Felix et al. [60] proposed a direct in situ transesterification in producing biodiesel form microalgae. Their results have shown that the global warming potential (GWP) impact of the direct in situ transesterification is approximately nine times lower when compared to a conventional microalgae biodiesel pathway. The transesterification tallied the highest impact contribution for the hazardous waste (HW and ozone depletion (ODP) due to the consumption of methanol as a catalyst to produce biodiesel).

For microalgae biodiesel production to be environmentally viable in remote areas, process-centric improvements need to be developed and implemented, such as cultivation using natural solar lighting and the employment of solar-assisted and solar dryers. Access to sophisticated equipment such as hydrothermal liquefaction or direct in situ transesterification may post challenges for remote community areas in producing biodiesel from microalgae. In addition, techno-economic analysis is one of the major considerations for the commercialization of algal biofuels. Recent works have focused on the techno-economic analysis of algal biofuels. Ahmad Ansari et al. [61] performed a techno-economic analysis on the commercialization of Scenedesmus obliquus growth through an integrated fish and biofuel production. Beckstrom et al. [62] utilized techno-economic analysis to evaluate the viability of the production of bioplastics and biofuels from microalgae. Rajesh Banu et al. [63] employed a techno-economic analysis to evaluate the operation of various configurations of a biorefinery to generate algal biofuels. One of the challenges of algal biofuels is competing commercially with currently available biofuels. In order for algal biofuels to be competitive, the processing requires co-production with a high-valued product such as the one mentioned in these studies.

## 5. Sustainability Model for Microalgae Biofuel Production

The previous sections demonstrated the feasibility and effectiveness of algal biofuels as an energy source. The current section will consider how algal biofuels can be successfully introduced in society as a replacement for conventional energy sources. The sustainability of algal biofuels depends on their continued adoption and use by the industry sector. This makes them a case study for technology diffusion, capacity expansion, and environmental policy. The system dynamics framework can incorporate these aspects in a single model. It also allows the sustainability of algal biofuels to be modeled and assessed, even without a large amount of data on existing successful implementations. The current section adopts this framework in identifying sustainable pathways for biofuel production.

Demand and supply are the primary actors in the diffusion of algal biofuels. The initial supply and the growth rate of supply are critical in facilitating the adoption of this energy source. Likewise, for any product, if the provider (of algal biofuels) fails to secure all demand, competitors (other energy

sources, including the current energy source being patronized by the industry) may service the demand. Figure 4 demonstrates this as a causal loop diagram, according to the system dynamics methodology. The consequence of this for biofuels is that the supply gap may discourage potential and current adopters. This would put to waste the existing biorefineries and resources invested in biofuel production.

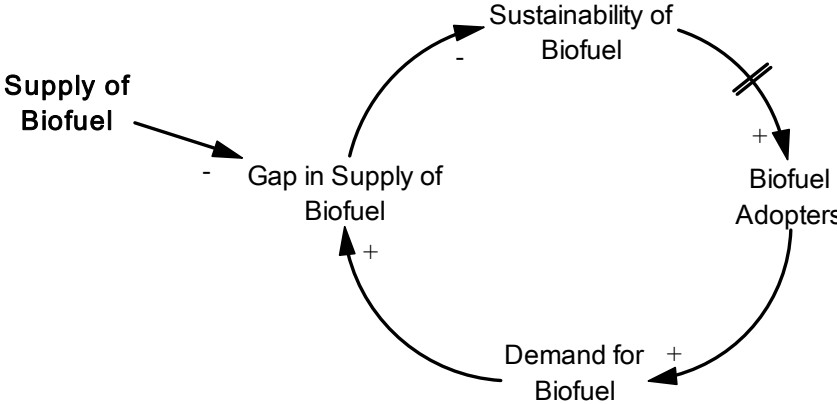

**Figure 4.** Demand and Supply Loop for Biofuels.

Increased demand can be met with capacity expansion. Capacity expansion may be driven by government initiatives. These initiatives may contribute a direct increase in capacity, through facilities funded by the government; or these may be in the form of incentives and penalties that encourage capacity expansion. These can give algal biofuels a strong entry into the industry sector. However, these initiatives also function as a short-term resource for capital expansion. The environmental issues, or interest in the technology, will diminish over time as the initiative begins to generate positive results (see Figure 5). This may be one way for demand to eventually exceed supply, in the long run.

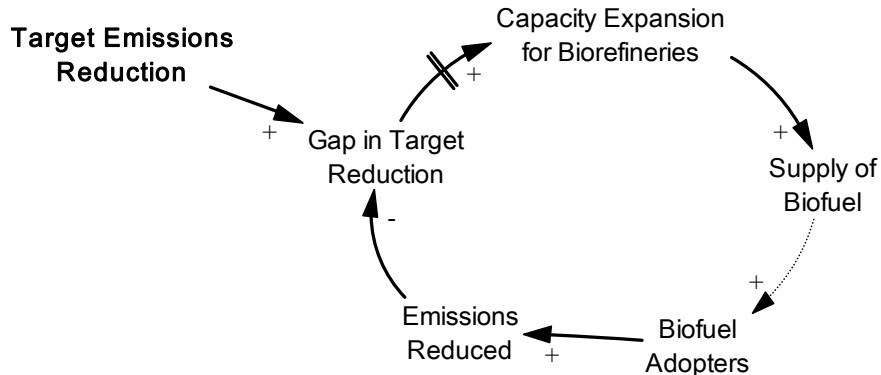

**Figure 5.** Environmental Incentives Loop for Biofuel Production.

Capacity expansion can also be driven by demand, through investors from the industry itself. As demand grows, algal biofuels proves to be a valuable industry in itself, and a fruitful investment. By basing capacity expansion on the funding from external investors, capacity grows continually, although whether or not demand can be met depends on the parameters of demand and supply growth (see Figure 6). In particular, energy yield and the rate of capacity expansion relative to demand, being the parameters which can be controlled, will determine the possibility of shortage, and consequently the sustainability of algal biofuels.

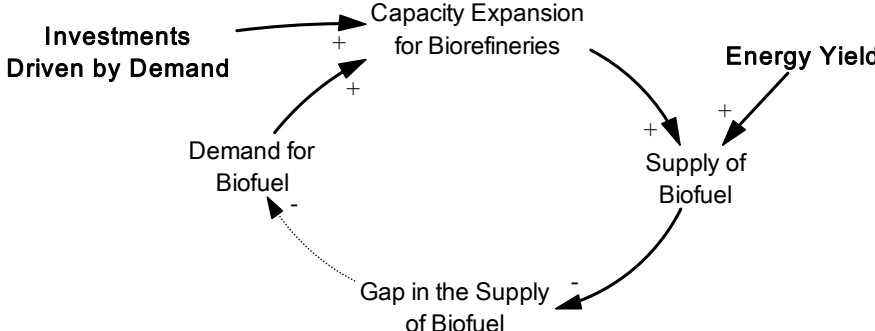

**Figure 6.** Investment Loop for Biofuel Production.

Aside from funding, other resources will be consumed in capacity expansion. This may create a competition for resources, as aggressive expansion will disrupt the balance of supply chains for these resources (see Figure 7). The efficient use of resources, with optimized energy yield, would be critical to ensuring that the rate of supply is able to grow in proportion to demand.

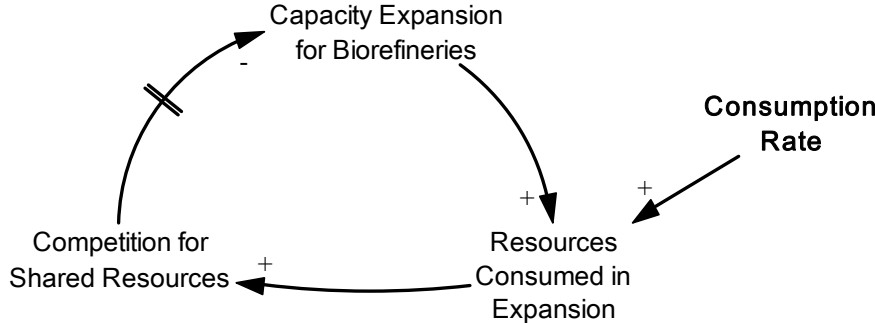

**Figure 7.** Resource Competition Loop for Biofuel Production.

It is important to note that the problems discussed in this section all grow in magnitude alongside demand. These can be addressed by streamlining capacity expansion. With this strategy, the following pathways toward sustainability are proposed:

1.  Channel funding toward research and development.

Research and development (R&D), targeted at increasing the energy yield of plants and all subsequent expansions, will be more efficient in ensuring that supply will be able to meet demand. Without R&D, the growth rate of supply will be constant, whereas with R&D, it increases over time. The higher growth rate is important in overcoming the barriers against expansion associated with demand (i.e., resource competition, supply gaps). A simulation, based on the system dynamics framework, demonstrates higher consistency and less chance for collapse in biofuel adoption (see Figure 8).

2.  Partner with the industry for expansion

Another means of increasing the growth rate of supply is through investments from the industry, as demonstrated in Figure 6. Investments are important as they provide a direct link between demand and capacity expansion, and hence a smoother growth in biofuel adoption in general (see Figure 9). The implications of the smoother curve are less barriers against entry for potential adopters, and also less openings for other energy sources which may fill the gap in supply. For greater efficiency, this solution may be coupled with further investments in R&D.

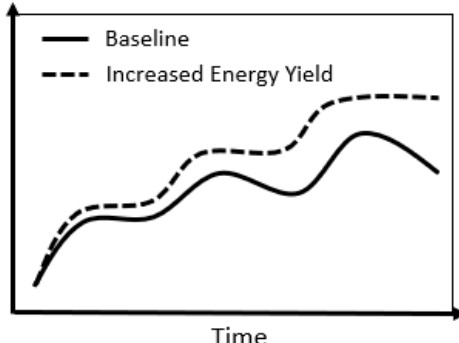

**Figure 8.** Simulated Impact of R&D on Biofuel Adoption.

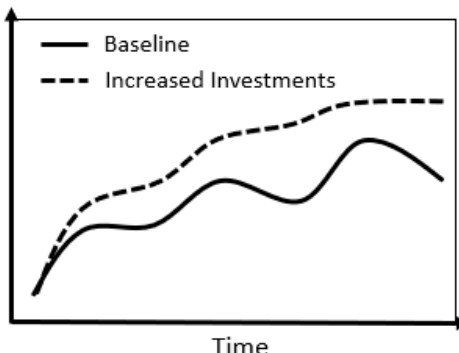

**Figure 9.** Simulated Impact of Industry Partners on Biofuel Adoption.

3. Scalable design for algal biorefineries

The outcomes achieved from investments can be replicated by changing the design of algal biorefineries. Generally, the bottleneck against expansion is the large amount of capital expenditure required to construct and operate algal biorefineries. By reinterpreting the increase in investment associated with demand to more capacity growth in proportion to the investments being made. This would also address resource competition to some extent. Land, one of the likeliest areas for resource competition, can be used more efficiently through biorefineries that are smaller in scale and strategically dispersed within an area.

**Author Contributions:** Conceptualization, A.B.C. and A.T.U.; methodology, A.T.U., W.-H.C. and J.-S.C.; software, P.M.L.C.; validation, W.-H.C. and J.-S.C.; formal analysis, A.T.U. and P.M.L.C.; investigation; A.B.C., A.T.U. and P.M.L.C.; resources, A.B.C.; writing—original draft preparation, P.M.L.C.; writing—review and editing, P.M.L.C.; project administration, A.B.C. All authors have read and agreed to the published version of the manuscript.

**Funding:** This research was supported by the Office of the Vice Chancellor of Research and Innovation of De La Salle University (DLSU), and the DLSU Science Foundation.

**Acknowledgments:** The authors acknowledge Engr. Roberto Louis Moran for the figures of the microalgal cultivation systems.

**Conflicts of Interest:** The authors declare no conflict of interest.

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
