# Peer review of "Biofuel from Microalgae: Sustainable Pathways"

_sustainability, doi:10.3390/su12198009_

Round 1
Reviewer 1 Report
In my opinion the submitted work entitled “Biofuel From Microalgae: Sustainable Pathways”, may be published almost in its present form (small editorial changes required).
This is an interesting and valuable job. The authors present and answer the most important questions related to biofuels.
I suggest only small corrections:
-Line 207 in the sentence “Coagulation/flocculation, centrifugation, and filtration are amongst the harvesting techniques that can be feasible for biofuel production”, the yellow-coloring/highlighting for the word ‘centrifugation’ should be removed;
-Line 264 missing Table 7 (“Table 7 shows a summary of these co-products and their purpose in the bio-refinery”);
-Line 291 in figure caption - “Table 1” should be replaced by “Table 3”;
-Line 295 as above, there is a reference to Table 7 in the text, which is not included in the paper;
-Line 339 - please explain the abbreviation “GWP”.
Author Response
Please see attachment for revised manuscript.
- Line 207 in the sentence “Coagulation/flocculation, centrifugation, and filtration are amongst the harvesting techniques that can be feasible for biofuel production”, the yellow-coloring/highlighting for the word ‘centrifugation’ should be removed;
The authors are grateful to the reviewer for noticing the error in formatting. This has been addressed in Line 225.
- Line 264 missing Table 7 (“Table 7 shows a summary of these co-products and their purpose in the bio-refinery”);
The authors are grateful to the reviewer for noticing that Table 7 had not been included in the manuscript. It has been included in the most recent version of the manuscript. It has been placed in Line 284.
- Line 291 in figure caption - “Table 1” should be replaced by “Table 3”;
The authors are grateful to the reviewer for noticing that the description in this line is more suited for Table 3 than Table 1. This can now be seen on Line 314.
- Line 295 as above, there is a reference to Table 7 in the text, which is not included in the paper;
Same response as (2).
- Line 339 - please explain the abbreviation “GWP”.
GWP is the abbreviation of Global Warming Potential. The full term and its abbreviation have been included in the paper, in Line 360.

Reviewer 2 Report
- extend table 1 to include other potential feedstock for biodiesel such as camelina, European rapeseed oil, and the USA canola.
- There are other technologies beyond what is presented in Fig 1 and 2. Please update the latest technology that it does exist in the market, and not necessarily exist in published papers.
- In table 4, most of the processes described are developed for woody biomass and not for microalgae. Therefore, table three should only describe various process based in microalgae, since the main title of this review is microalgae. Besides, for each technology you presented, there must be a citation from recent publication.
- The author needs to create a special section discussing the economic challenges and discuss recent economic calculation date, for example, capital investment and operation cost with other studies.
Here are some important work that help you address the above comments. Please refer to these articles and cite them when it is possible:
- Yang, Changyan, et al. "Pyrolysis of microalgae: A critical review." Fuel Processing Technology 186 (2019): 53-72.
- Goh, Brandon Han Hoe, et al. "Sustainability of direct biodiesel synthesis from microalgae biomass: A critical review." Renewable and Sustainable Energy Reviews 107 (2019): 59-74.
- Mohammad, Balsam T., et al. "Production of multiple biofuels from whole Camelina material: a renewable energy crop." BioResources 13.3 (2018): 4870-4883.
- Spolaore, Pauline, et al. "Commercial applications of microalgae." Journal of bioscience and bioengineering 101.2 (2006): 87-96.
- Acién, F. G., J. M. Fernández, and E. Molina-Grima. "Economics of microalgae biomass production." Biofuels from Algae. Elsevier, 2014. 313-325.
- Treatment and kinetic study of cyanobacterial toxin by ozone, Journal of Environmental Science and Health - Part A Toxic/Hazardous Substances and Environmental Engineering, Volume 45, Issue 6, May 2010, Pages 719-731
Author Response
Please see attachment for revised manuscript.
- extend table 1 to include other potential feedstock for biodiesel such as camelina, European rapeseed oil, and the USA canola.
Reply:
The authors acknowledged the suggestion of the reviewer by improving Table 1. Table 1 was revised to include the details of camelina, and canola/rapeseed biomass feedstock.
- There are other technologies beyond what is presented in Fig 1 and 2. Please update the latest technology that it does exist in the market, and not necessarily exist in published papers.
Reply:
The authors are grateful for the comments of the reviewer and the suggestion to add a discussion on other cultivation technologies. As a response, the authors have added a paragraph in section 2.1 to discuss other cultivation technologies. The paragraph added is shown below.
“New approach in the cultivation and growth of microalgae were developed and designed. Recently, an indoor hybrid helical-tubular photobioreactor was proposed by Hashemi et al. (2020) for beta-carotene production Dunaliella salina cells under salt stress condition. A newly developed microgravity-capable membrane raceway photobioreactor for Chlorella vulgaris SAG 211-12 was introduced by Helisch et al. (2020) for life support in space. On the other hand, Khalekuzzaman et al. (2019) developed a hybrid anaerobic baffled reactor and photobioreactor for a simplified method of algal biofuel production. It is important for algal biofuel production to be simple and cost-effective to make it competitive with commercially available biofuels.”
References:
Hashemi, A., Moslemi, M., Pajoum Shariati, F., Delavari Amrei, H., (2020). Beta-carotene production within Dunaliella salina cells under salt stress condition in an indoor hybrid helical-tubular photobioreactor. Canadian Journal of Chemical Engineering 98(1): 69-74.
Helisch, H., Keppler, J., Detrell, G., Belz, S., Ewald, R., Fasoulas, S., Heyer, A.G., (2020). High density long-term cultivation of Chlorella vulgaris SAG 211-12 in a novel microgravity-capable membrane raceway photobioreactor for future bioregenerative life support in SPACE. Life Sciences in Space Research 24: 91-107.
Khalekuzzaman, M., Alamgir, M., Islam, M.B., Hasan, M., (2019). A simplistic approach of algal biofuels production from wastewater using a Hybrid Anaerobic Baffled Reactor and Photobioreactor (HABR-PBR) System. PLoS ONE 14(12): Article number e0225458.
- In table 4, most of the processes described are developed for woody biomass and not for microalgae. Therefore, table three should only describe various process based in microalgae, since the main title of this review is microalgae. Besides, for each technology you presented, there must be a citation from recent publication.
Reply:
The authors acknowledged the suggestion of the reviewer. However, the technologies outlined in Table 4 are matured technologies which can be used in any biomass feedstock. To further qualify the technologies for the production of algal biofuels, there are technologies that can leapfrog the processing of microalgae to biofuels. For instance, hydrothermal liquefaction was found to be effective for microalgal biomass for the following reasons. The microalgal biomass inherently contains high moisture which requires high energy consumption from drying. Hydrothermal liquefaction enables the processing of highly moist microalgal biomass to produce high yield biofuels. It bypass the need for the drying process, hence, reducing the energy requirement for the production of algal biomass.
To capture this in the text, a discussion was added on hydrothermal liquefaction in section 2.3. The added paragraph is shown as follows.
“Hydrothermal liquefaction technology shows promising results in the production of algal biofuels as it enables the processing of wet microalgae biomass to produce biofuels. Recent works on hydrothermal liquefaction for algal biofuels are discussed as follows. Devi and Parthiban (2020)” proposed the application of hydrothermal liquefaction on microalgae Nostoc ellipsosporum cultivated in municipal wastewater to generate high bio-oil yield. Dandamudi et al. (2020) applied hydrothermal liquefaction on Cyanidioschyzon merolae and Salicornia bigelovii Torr. for the production of high-valued biofuel intermediates. Arun et al. (2020) used hydrothermal liquefaction on Scenedesmus obliquus to produce biohydrogen. The employment of hydrothermal liquefaction on microalgae biomass to biofuel production provides positive benefit for the cost-effectiveness, value-addition, and environmental impact of the algal biofuels.”
References:
Devi, T.E., Parthiban, R., (2020). Hydrothermal liquefaction of Nostoc ellipsosporum biomass grown in municipal wastewater under optimized conditions for bio-oil production. Canadian Journal of Chemical Engineering 98(1): 69-74. Bioresource Technology 316, Article number 123943.
Dandamudi, K.P.R., Muhammed Luboowa, K., Laideson, M., Murdock, T., Seger, M., McGowen, J., Lammers, P.J., Deng, S., (2020). Hydrothermal liquefaction of Cyanidioschyzon merolae and Salicornia bigelovii Torr.: The interaction effect on product distribution and chemistry. Fuel 277, Article number 118146.
Arun, J., Gopinath, K.P., SundarRajan, P., Malolan, R., Adithya, S., Sai Jayaraman, R., Srinivaasan Ajay, P., (2020). Hydrothermal liquefaction of Scenedesmus obliquus using a novel catalyst derived from clam shells: Solid residue as catalyst for hydrogen production. Bioresource Technology 310, Article number 123443.
- The author needs to create a special section discussing the economic challenges and discuss recent economic calculation date, for example, capital investment and operation cost with other studies.
Reply:
The authors are thankful for the suggestion of the review to include the techno-economic discussion of algal biofuels. However, the main objective of the study showcased the sustainable pathways of biofuel from microalgae. The study only focuses mainly on the environmental assessment using sustainability models. Thus, the techno-economic analysis of the algal biofuels is out of the scope of the study. However, the authors have added the following paragraph to fill-in the discussion on the economic challenges of algal biofuel production.
“Techno-economic analysis is one of the major consideration for the commercialization of algal biofuels. Recent works have focused on the techno-economic analysis of algal biofuels. Ahmad Ansari et al. (2020) performed a techno-economic analysis on the commercialization of Scenedesmus obliquus growth through an integrated fish and biofuel production. Beckstrom et al. (2020) utilized techno-economic analysis to evaluate the viability of production of bioplastics and biofuels from microalgae. Rajesh Banu et al. (2020) employed techno-economic analysis to evaluate the operation of various configuration of a biorefinery to generate algal biofuels. One of the challenges of algal biofuels is competing commercially with currently available biofuels. In order for algal biofuels to be competitive, the processing requires co-production with a high-valued product such as mentioned in these studies.”
References:
Ahmad Ansari, F., Nasr, M., Guldhe, A., Kumar Gupta, S., Rawat, I., Bux, F., (2020). Techno-economic feasibility of algal aquaculture via fish and biodiesel production pathways: A commercial-scale application. Science of the Total Environment 704, , Article number 135259.
Beckstrom, B.D., Wilson, M.H., Crocker, M., Quinn, J.C., (2020). Bioplastic feedstock production from microalgae with fuel co-products: A techno-economic and life cycle impact assessment. Algal Research 46, Article number 101769.
Rajesh Banu, J., Preethia, Kavitha, S., Gunasekaran, M., Kumar, G., (2020). Microalgae based biorefinery promoting circular bioeconomy-techno economic and life-cycle analysis. Bioresource Technology 302, Article number 122822.

Round 2
Reviewer 2 Report
The authors responded seriously to all my comments. The final decision is to accept.